# Interplay Impact of Exogenous Application of Abscisic Acid (ABA) and Brassinosteroids (BRs) in Rice Growth, Physiology, and Resistance under Sodium Chloride Stress

**DOI:** 10.3390/life13020498

**Published:** 2023-02-10

**Authors:** Sajid Hussain, Satyabrata Nanda, Muhammad Ashraf, Ali Raza Siddiqui, Sajid Masood, Maqsood Ahmed Khaskheli, Muhammad Suleman, Lianfeng Zhu, Chunquan Zhu, Xiaochuang Cao, Yali Kong, Qianyu Jin, Junhua Zhang

**Affiliations:** 1State Key Laboratory of Rice Biology, China National Rice Research Institute, Hangzhou 310006, China; 2Soil and Water Testing Laboratory, Marketing Division, Pak Arab Fertilizer Limited, Multan 66000, Pakistan; 3MS Swaminathan School of Agriculture, Centurion University of Technology and Management, Paralakhemundi 761211, India; 4Department of Soil Science, Bahauddin Zakariya University, Multan 60700, Pakistan; 5Department of Soil Science, Faculty of Agricultural Sciences, Quid-E-Azam Campus, University of the Punjab, Lahore 54590, Pakistan; 6Department of Plant Pathology, Agricultural College, Guizhou University, Guiyang 550001, China

**Keywords:** rice, phytohormone, salt stress, photosynthesis, SPAD value, grain yield

## Abstract

The hormonal imbalances, including abscisic acid (ABA) and brassinosteroid (BR) levels, caused by salinity constitute a key factor in hindering spikelet development in rice and in reducing rice yield. However, the effects of ABA and BRs on spikelet development in plants subjected to salinity stress have been explored to only a limited extent. In this research, the effect of ABA and BRs on rice growth characteristics and the development of spikelets under different salinity levels were investigated. The rice seedlings were subjected to three different salt stress levels: 0.0875 dS m^−1^ (Control, CK), low salt stress (1.878 dS m^−1^, LS), and heavy salt stress (4.09 dS m^−1^, HS). Additionally, independent (ABA or BR) and combined (ABA+BR) exogenous treatments of ABA (at 0 and 25 μM concentration) and BR (at 0 and 5 μM concentration) onto the rice seedlings were performed. The results showed that the exogenous application of ABA, BRs, and ABA+BRs triggered changes in physiological and agronomic characteristics, including photosynthesis rate (Pn), SPAD value, pollen viability, 1000-grain weight (g), and rice grain yield per plant. In addition, spikelet sterility under different salt stress levels (CK, LS, and HS) was decreased significantly through the use of both the single phytohormone and the cocktail, as compared to the controls. The outcome of this study reveals new insights about rice spikelet development in plants subjected to salt stress and the effects on this of ABA and BR. Additionally, it provides information on the use of plant hormones to improve rice yield under salt stress and on the enhancement of effective utilization of salt-affected soils.

## 1. Introduction

The rapid increase in the human population demands a 70% rise in the food supply [1]. A rapid surge in the production of cereals, especially rice, is required to meet such enormous demands. Rice (*Oryza sativa* L.) is among the key cereal crops and a staple food for almost 50% of the world’s population [2,3]. However, soil salt content and salt stress have emerged as major threats to global rice agriculture. The extent of loss of rice yield due to salt stress depends on several factors, including stress duration, day length, development stage of the rice, and so on [4]. Salt stress is dominated by sodium (Na^+^) and chloride (Cl^−^) ions, which affect the rice’s growth and development by creating ionic, osmotic, and oxidative imbalances [5]. Higher salt levels raise leaf toxicity in rice plants, which results in senescence of leaves and reduced leaf area, which ultimately reduces the photosynthesis rate [6,7,8]. Furthermore, low rice yield is largely related to a reduction in tiller numbers and spikelet sterility due to salt stress effects. Spikelet sterility due to salt stress has become an alarming issue in rice agriculture around the world. As a result, the seed set can be decreased by about 38%, even when the rice plants experience lower salt levels, such as 10 mM NaCl [9]. Previously published studies have reported that spikelet sterility and a reduction in seed setting rate were observed due to the decreased translocation of soluble carbohydrates to superior and inferior spikelets, more accumulation of sodium (Na^+^), and less accumulation of potassium (K^+^) in all floral parts of the spikelet, causing a reduction in starch synthesis in rice grains, which is key for grain development [10,11]. Additionally, the role of phytohormones and their interplay is vital in regulating spikelet development and grain sensitivity in rice [12].

Phytohormones as chemical messengers play crucial roles in plant stress responses with respect to plant growth and development [13]. Under salt stress, phytohormones can aid in important plant adaptations to the stress by facilitating nutrient balance, a translocation between source and sink, and plant growth and development [14,15]. In rice, ethylene (ET), abscisic acid (ABA), brassinosteroids (BRs), gibberellins (GAs), cytokinins (CKs), auxins (IAA), jasmonic acid (JA), and salicylic acid are the key phytohormones that modulate numerous rice physiological processes and stress responses [16]. Under salt stress, plants try to mitigate the stress effects by regulating the ABA levels. Rapid ABA accumulation causes stomatal closure, which ensures a reduction in water loss via transpiration [17,18]. A high ABA concentration in leaf tissues helps in salt adaptation by osmotic adjustment, altering stomatal activity, and improving stress protein production [17]. On the other hand, the BRs, a class of polyhydroxy steroidal phytohormones, have a significant influence on rice physiological responses [19]. BRs mitigate the negative impact of salt stress on physiological, molecular, and biochemical mechanisms, thus contributing to a plant’s salinity tolerance ability [20]. For instance, the exogenous application of 24-epibrassinolides (24-epiBL) increased growth and mitigated salt effects in treated pea plants as compared to controls [21,22]. Similarly, the application of 5 μM epiBL onto the *Phaseolus vulgaris* plants detoxified the effects of salt stress and improved pea growth, green pod yield, and pod protein content significantly [21,23]. Moreover, supplementation of 24-epiBL could play pivotal roles in stress response and phytohormonal dynamics in rice plants under salt stress [22]. In addition, several efforts have been made to minimize the negative impact of salt stress and regulate the hormone balance in rice plants by using the exogenous application of hormones and some stress hormone inhibitors [16,22].

For instance, the analogs of BRs, DI-31 or Biobras-16 (BB16), could be a better option for exogenous application in rice plants. DI-31 or BB16 possess BS-like activity by having a spiroketalic ring instead of the typical BR side chain [24]. In addition, the ABA and BR crosstalk is vital in regulating several physiological processes in rice. For example, the antagonistic interaction between ABA and BR is well studied in rice. Although these interactions were thought to start late, some findings reported the early crosstalk of ABA and BR at the formation of the BRI1–BAK1 receptor complex [25,26]. The fine-tuned interactions of ABA and BR in rice offer a mechanism to switch between growth and stress response depending on the BR and ABA levels [27]. However, the effects of exogenous application of these analogs, either independently or synergistically with abscisic acid, to reveal their roles in rice growth and resistance to salt stress, have not been studied previously. Therefore, in the current study, we investigated the impact of independent and combined exogenous treatment with D-31 (BB16) and ABA on rice growth, spikelet development, physiology, hormone production, and salt resistance.

## 2. Materials and Methods

### 2.1. Growth Conditions

A pot culture experiment was conducted in a rainout shelter during the rice-growing season (May–November 2019) at the China National Rice Research Institute (39°4′49′′ N, 119°56′11′′ E), Zhejiang Province, China. One rice cultivar, Liangyoupeijiu (LYP9, indica), with high germination (>95%) was used as plant material. The soil used in this trial was clay loam with EC 0.086 dS m^−1^ as control (CK), and the pH value was 5.95. Normal paddy field-grown seedlings were transplanted after 30 days of sowing into pots, and each pot (30 × 40 cm) had 30 kg of air-dried clay loam soil. Six seedlings per pot were used.

### 2.2. Treatment Plan

Abscisic acid (ABA) and analogs of brassinosteroids (BRs), BB16 or DI-31, were used as stress regulators in this experiment. The effects of ABA and BRs on rice growth characteristics and the development of spikelets were explored under salinity. Previously developed saline-soil pots were used for salinity stress induced by sodium chloride (NaCl). The electrical conductivity (EC) values of the available soil pots were as 0.0875 (control, CK), ~2 dS/m (1.878 dS m^−1^, low salinity stress, LS), ~4 dS/m (4.09 dS m^−1^ (heavy salinity stress, HS). The hormones treatments were as follows: ABA (0 and 25 μM), BRs (0 and 5 μM), and a combination of hormones (ABA + BRs, 0, 25 μM + 5 μM). The exogenous hormones were applied at the time of transplanting rice seedlings and at the rice booting stage. The time frame for rice growth data is given in Table 1.

For the phytohormone application, a complete randomized design (CRD) method was applied. The experiment design was divided into four sections: (a) no hormone application at each salinity level, (b) ABA application at each salinity level, (c) BR application at each salinity level, and (d) ABA+BRs at each salinity level. Each treatment had three replicates and six sampling times. For the whole experiment, a total of 216 pots were used: 3 salinity levels * 2(ABA levels) * 2(BRs levels) * 2(ABA+BRs levels) * 1 rice cultivar * 3 replicates * 6 sampling times.

### 2.3. Hormone Application

ABA and BRs were obtained from CID Technologies Inc. (Brampton, Ontario, Canada). Based on pre-experiment results, the concentrations were selected as ABA (0 and 5 μM), BRs (0 and 5 μM), and ABA+BRs (0, 25 μM + 5 μM). The required quantity of ABA or BRs was prepared in 5 mL of ethanol in a 100 mL volumetric flask and the final volume was made up using double-distilled water containing Tween-200 (containing 0.1% *v*/*v*) as sticky material. The working concentrations of ABA (25 μM) and BRs (5 μM) were prepared by diluting stock with double-distilled water. Plants were sprayed twice a week with the hormones (ABA and BRs) after 10 days of transplanting, and at the booting stage. Sprays of the solvent (ethanol + distilled water) were used on the control plants.

### 2.4. Fertilizer Application

Urea as the nitrogen (N) source, superphosphate as the P_2_O_5_ source, and potassium sulfate as the K_2_O source were used to supply NPK to the soil. Urea (46% N) was applied at a rate of 5.24 g per pot in three doses: 50% as basal dose, 30% at the tillering stage, and 20% at the rice booting stage. Superphosphate was applied at a rate of 9.04 g/pot as a basal dose (100%). Potassium sulfate (K_2_SO_4_) was applied at a rate of 4.02 g/pot in two doses: 50% as basal and 50% at the booting stage.

### 2.5. Measurement of Rice Growth Characteristics

Three plants from each pot were taken randomly and measured for plant height, leaf area, panicle length, and total dry biomass. For the analysis of hormones produced in the leaves, flag leaves were taken at the full heading stage. In addition, for the analysis of pollen viability spikelets were taken after the anthesis stage. For the analysis of spikelet sterility, spikelet samples were taken from each treatment at the full heading stage. All samples were taken in three replicates.

### 2.6. Photosynthesis Parameters and Leaf Area

The plant physiological parameters, such as photosynthesis rate (Pn), stomatal conductance, and water transpiration rate of the flag leaf were recorded after the application of hormones. The data were collected in daylight between 8:30 a.m. and 11:30 a.m. using a portable photosynthesis system (LI-6400XT) (LI-COR Inc., Portland, OR, USA). While recording the data, the concentration of carbon dioxide (CO_2_) in the leaf chamber was 400 µmol mol^−1^, light intensity was 1500 µmol m^−2^ S^−1^, and the temperature was 30 °C. 

SPAD value (chlorophyll content) of the flag leaf was measured after a 15-day interval followed by hormone application using a chlorophyll meter (SPAD-502 plus, Tiangen, China). 

The leaf area (LA) of the plant was measured using a leaf area meter (LI-3100 Core Area Meter, Lincoln, NE, USA).

### 2.7. Quantification of the Hormones Produced in the Rice Flag Leaf

In this experiment, the hormones abscisic acid (ABA), gibberellic acid (GA3), brassinosteroids (BRs), jasmonic acid (JA), and indole-3-acetic acid (auxin, IAA) were quantified. For this purpose, about 0.5 to 1.0 g flag leaf sample was taken. Samples were stored immediately in aluminum foil, snap-frozen by putting the samples into liquid nitrogen, and kept at −80 °C. Two grams of the samples were homogenized on an ice bath and transferred into 10 mL tubes. The tubes were kept at 4 °C for 1 h. Subsequently, the samples were centrifuged for 8 min at 3500 rpm by high-performance fixed-angle rotor for 4 h at 4 °C, and the supernatant was collected. The precipitate was mixed well with 1ml of extract, placed at 4 °C, recentrifuged for 1 h, and the volume was recorded. The supernatant was applied to a C-18 solid-phase extraction column.

The measuring steps are as follows: the column was balanced with 1 mL of 80% methanol and the samples were loaded after washing with 5 mL of 100% methanol; the column was then washed again with 5 mL of 100% ether and 100% methanol (5 mL), completing the cycle. After that, methanol was removed from the extract, and samples were shifted to 10 mL plastic centrifuge tubes and dried with nitrogen. The sample was then diluted with liquid to a fixed volume, i.e., 2 mL sample diluted with 1 mL distilled water. 

For ELISA-based quantification, the samples and standards were taken and different concentration solutions were made. For IAA and ABA, the concentration of the standard solutions was made to 50 ng/mL. Similarly, for GA and BR, the concentration was made to 2 ng/mL. For JA, the standard solution concentration was made to 10 ng mL^−1^. Eight different dilutions were made for the test samples and used for constructing the standard curve. The different dilutions of the test samples and the standards were loaded into a 96-well ELISA plate and the concentrations were recorded. The experiment was repeated thrice to deduce the concentrations. The final concentration was measured in ng g^−1^ FW.

### 2.8. Measurement of Pollen Viability and Spikelet Sterility

At the rice anthesis stage, the spikelets bearing anthers were taken. The staining process for mature pollen grains used potassium iodide (KI) and iodine solution (I_2_) with a concentration of 0.5 g of I_2_ and 2 g of KI in 100 mL of H_2_O. The matured pollen grains from the anthers were stained by adding a drop of potassium iodide/iodine solution (KI/I_2_) onto a glass slide and imaged 100× under a light microscope (DM4000B, Leica, Wetzlar, Germany). Finally, pollen grains were categorized as fertile or unfertile on the basis of their color. 

Spikelet sterility was measured by visual observation with a stereoscope and the images were analyzed manually to access the extent of sterility in the spikelet.

### 2.9. 1000-Grain Weight and Rice Grain Yield Per Plant

Three plant samples were harvested at maturity for measurement of yield and 1000-grain weight. Fully grown rice plants were harvested carefully from the greenhouse and were incubated at room temperature until the samples reached a constant weight, after which the grain yield and 1000-grain weight were calculated.

### 2.10. Statistical Analysis

The complete randomized design (CRD) was used in this study. Statistical analysis was performed by comparing pairs of mean values by least (one-way ANOVA) significant difference (LSD) at the level of 5% significance using the IBM SPSS statistics 19.0 software package.

## 3. Results

### 3.1. Effects of Exogenous Hormonal Treatments on Rice Agronomic Traits at Full Heading Stage under Salinity Stress

Salinity stress reduced the plant height, leaf area, panicle length, and total plant dry matter at the full heading stage under various salinity stresses induced by sodium chloride (NaCl) salt levels (CK, LS, and HS) as compared to the hormonal treatments. In this experiment, under no hormone application, the plant height (11.03% and 50.5%), leaf area per plant (0.2% and 12.3%), panicle length (1.5% and 26.1%), and total plant dry matter (2.6 and 46.7%) of LPY9 decreased in low salt stress (LS) and high salt stress (HS) compared to control (CK). In the case of exogenous application of BRs, the plant height decreased by 35.6% (LS) and increased by 13.2% (HS) as compared to no salinity stress (CK). Similarly, the leaf area per plant (9.7 and 39.7%), panicle length (8.3 and 35.2%), and total plant dry matter (7.0 and 29.2%) also decreased under low salinity (LS) and high salinity (HS) as compared to CK (Figure 1). Conversely, the application of ABA improved the plant height by 44.9% (LS) and 42.5% (HS), leaf area per plant by 0.2 (LS) and 12.3% (HS), decreased the panicle length by 8% (LS) and 32.4% (HS), and decreased total plant dry matter by 19.6% (LS) and 24.4% (HS) as compared to no hormone treatment under CK, LS, and HS, respectively. ABA+BR application had similar but nonsignificant effects: plant height increased by 2.65% (LS) and decreased by 38.6% (HS), leaf area decreased by 5.9% (LS) and 13.1% (HS), panicle length decreased by 0.8% (LS) and 38.2% (HS), and total plant dry matter decreased by 16% (LS) and 36% (HS) as compared to CK under hormone treatments (Figure 1).

### 3.2. Effects of Exogenous Hormones on Photosynthesis Attributes of Rice Flag Leaf under Salinity

Salinity affects the physiological traits of the rice plant. In the current study, the photosynthesis attributes were altered by increasing salinity levels (Table 2). The net photosynthesis rate (Pn) value in rice flag leaves was decreased by 20.7% in LS and 20.3% in HS salinity levels, respectively, compared to no salinity stress (CK) with no hormone treatment. However, a difference was observed after the exogenous applications of hormones (BRs, ABA, ABA+BRs) compared to no hormone treatment. The Pn in rice flag leaf was reduced by 17.44% under LS with BR application, by 24.24% (LS) with ABA, and 18.19% (LS) with ABA+BRs treatment compared to control (CK) with the application of hormonal treatments, i.e., BRs, ABA, ABA+BRs. Similarly, at high salt stress (HS), the Pn value was decreased by 22.11% with BRs and 36.36% with ABA, and significantly decreased by 26.03% with ABA+BR treatment compared to CK under each hormonal application (BRs, ABA, ABA+BRs; Table 2).

When comparing hormone application with no hormone application at all salt stress levels, the results show that no change was found in Pn with CK (BRs) compared to CK with no hormone application. However, under CK (ABA) and CK (ABA+BRs), the Pn value was increased by 13.1% and 10.4% compared with CK (no hormone). On the other hand, the response was different in terms of LS (no hormones) compared to LS (with hormones). The Pn value in LS (BRs) and LS (ABA) was decreased by 7.8% and 2.6%, and in LS (ABA+BRs) it was increased by 0.73% compared to LS (no hormones). A similar trend was observed in terms of HS (hormones) compared to HS (no hormones). The Pn value in HS (BRs) and HS (ABA) was decreased by 2.91% and 8.7%, respectively, and in HS (ABA+BRs), it was increased by 2.83% compared to HS (no hormones). These results indicate that with increasing salt stress level (HS), the combined application of ABA+BRs had a positive impact compared to HS (no hormones). Similarly, hormonal applications improved the Pn in CK compared to CK receiving no hormones.

Without hormone application, the SPAD value was decreased by 7.3% (LS) and 12.2% (HS) compared to no salinity stress (CK). However, a difference was observed after the exogenous applications of hormones. The application of BRs, ABA, and ABA+BRs improved the SPAD value by 0.5% (BRs), decreased it by 3.5% (ABA), and increased it by 1.5% (ABA+BRs) under LS, compared to CK under each hormonal treatment. The application of BRs, ABA, and ABA+BRs improved the SPAD value by 10.5% (BRs), decreased it by 1.5% (ABA), and increased it by 1.2% (ABA + BRs) under HS compared to CK under each hormonal treatment, respectively (Table 2). Overall, applying hormones improved the SPAD value of rice flag leaf in BR and ABA+BR treatments compared to the value without hormones.

Similarly, the stomatal conductance (gs) and transpiration rate (Tr) were studied under the influence of hormone application in plants subjected to salinity stress. The results revealed that gs and Tr improved with the application of ABA+BRs and ABA alone compared to no hormone application at all salinity stress levels. These results suggest that hormone homeostasis is crucial for high plant growth (Table 2).

### 3.3. Effects of ABA and BRs on Phytohormonal Dynamics under Salinity

The phytohormones showed significant variations in LYP9 under the influence of salinity stress and exogenous application of ABA and BRs. Application of ABA, BRs, and ABA+BRs showed an interesting response at all salinity stress levels (CK, LS, and HS) (Figure 2a–e). BR production was higher in CK with no hormone treatment and with ABA+BR treatment, whereas BRs showed possible antagonistic effects on the ABA production in rice flag leaves at all salinity stress levels (Figure 2a). ABA production was the highest under HS levels with BR treatment, and ABA production was stable at all salinity stress levels with ABA+BR treatment. The trends of GA3 production in rice flag leaves were similar among all treatments. However, the GA3 production response was different with BR treatment at LS compared to CK and HS levels (Figure 2c). Similarly, the production response of JA was nonsignificant with no hormone treatment. When treated with BRs, JA production showed an increasing trend with the increase in salinity level. Furthermore, JA production was higher with ABA+BR treatment (Figure 2d). In addition, JA production showed an increasing trend with BR and ABA treatment with the increasing salinity levels, but the trend was possibly antagonistic in no hormone treatment compared to ABA+BRs at each salinity level (Figure 2e). The results revealed that most of the analyzed hormones showed increased responses with BR treatment. However, treatment with ABA+BRs gives a clue that the application of both hormones in mixed form maintains homeostasis in the production of phytohormones (Figure 2a–e). 

### 3.4. Pollen Viability and Spikelet Sterility 

Pollen viability during grain development plays a significant role in the seed setting rate. Furthermore, poor pollen viability causes spikelet sterility. In this study, the negative effects on pollen viability and spikelet sterility in LYP9 were observed under different salinity stress levels (Figure 3). The exogenous application of BRs, ABA, and ABA+BRs improved pollen viability under salinity stress conditions (Figure 3), especially BRs at all salinity levels (CK, LS, and HS) compared to no hormone treatment. The comparison between the two hormones (BRs and ABA) revealed that ABA treatment had more viable pollen than BRs at CK and LS levels, whereas ABA+BRs showed more pollen viability than other hormone treatments at the HS salinity level (Figure 3). In contrast, the BR and ABA+BR applications showed better results with regard to reducing spikelet sterility. These results revealed that high pollen viability and low spikelet sterility play an important role in the grain setting rate and rice yield, even under unfavorable conditions such as salinity stress.

### 3.5. Effects of ABA and BRs on Rice Grain Weight under Salinity Stress 

The grain development process is the key to estimating grain yield. In this experiment, the grain weight improved with the exogenous application of hormones (ABA, BRs, ABA+BRs) as compared to the no-hormone-treated plants at the increased salinity levels (Figure 4). After ten days of reaching the full heading stage of the rice plant, the inferior and superior grain weights in the ABA, BR, and ABA+BR treatments were more or less similar to the grain weight of the no-hormone-treated plants at various salinity stress levels. However, the grain weight in superior spikelets was increased with the application of BRs, as compared to the no-hormone-treated plants at all salinity levels Figure 4(1a–1c). A similar trend was observed with the application of ABA when compared to the no-hormone-treated plants at each salinity stress level Figure 4(2a–2c). Moreover, the grain weight was significantly improved with the application of ABA+BRs at all salinity levels, as compared to the no-hormone-treated plants Figure 4(3a–3c). This positive effect of hormones on grain development may be due to higher photosynthesis attributes (Pn and SPAD value) in the LYP9 rice cultivar.

### 3.6. Effects of ABA and BRs on Rice Grain Yield and Its Attributes under Salinity Stress 

The ultimate effects of salinity stress are reductions in yield and its components. In the present study, a significant reduction in grain yield per plant and 1000-grain weight was observed under salinity stress. The 1000-grain weight and grain yield per plant were altered by the increasing salinity levels (Table 3). In the no-hormone-treated plants, the 1000-grain weight was decreased by 5.6% (LS) and 61.5% (HS) compared to no salinity stress (CK). However, with hormonal treatments (ABA and BRs) a difference was observed. The application of BRs, ABA, and ABA+BRs increased the 1000-grain weight under low salinity stress (LS) by 4.8% with BRs, 4.6% with ABA, and 2.9% with ABA+BRs, rising to 16.7% with BRs, 27.1% with ABA, and 39.1% with ABA+BRs under HS compared to CK (Table 3). Overall, hormone application improved the 1000-grain weight compared to the weight without hormones.

The grain yield per plant was decreased by 4% (LS) and 80% (HS) compared to no salinity stress and no hormone application. However, a difference was observed after the exogenous application of hormones. The application of BRs, ABA, and ABA+BRs improved the grain yield per plant by 5.9% (BRs), 11.8% (ABA), and 8.5% (ABA+BRs) under LS, and decreased it by 63.1% (BRs), 60.8% (ABA), and 71.7% (ABA+BRs) under HS compared to CK (Table 3). Overall, hormone applications improved the grain yield per plant in the BRs group compared to no hormones under CK.

## 4. Discussion

Salinity stress has multiple effects on rice plant physiology and growth. These effects can vary depending on the rice growth stage and level of salinity. In addition, rice phytohormones play pivotal roles in regulating salinity tolerance by influencing several key physiological processes and signaling [16]. In the present study, subjection to salinity stress (LS and HS) significantly affected rice growth and other agronomic parameters. A significant reduction in the net photosynthesis rate (Pn) and SPAD value were recorded under salinity stress, which implied an effect of salinity on photosynthesis and stomatal closure. Owing to the closure of stomata and the inhibition of water conductance in mesophyll cells caused by osmotic stress under salinity stress, the leaf turgor, and signaling could have been impaired [28]. In contrast, the exogenous application of ABA, BRs, and ABA+BRs improved the Pn and SPAD values of rice flag leaves at different salinity levels. Additionally, salinity reduced plant height, leaf area, panicle length, and total plant dry biomass of the rice plants. The literature suggests that many plant growth traits are affected by the hormonal imbalance caused by salinity [29]. The reduction in the rice plant parameters might be due to a hormonal imbalance, such as biosynthesis and redistribution of ABA that regulates the stomatal closure, osmotic stress, CO_2_ assimilation, and biomass production [30]. In our study, raised salinity levels led to the disequilibrium of phytohormones, including ABA, BRs, JA, GA3, and IAA, in rice (Figure 4). Rice hormonal homeostasis can be altered by salinity stress [16]. Conversely, the exogenous application of ABA, BRs, and ABA+BRs significantly improved the performance of rice plants under salinity. These changes could have been generated by hormonal regulation returning to homeostatic levels. The exogenous application of ABA plays an important role in tissues and leaves, not only by altering the stomatal activity but also by improving the production of stress proteins and by salinity adaptation via osmotic adjustment [31]. BRs, however, confer salinity tolerance to plants by mitigating the negative effects of salt on the physiological, biochemical, and molecular processes in plants [20]. Exogenous application of BRs offered tolerance to salinity by altering stress responses in rice (Ashraf et al., 2010) [32]. Both ABA and BRs have an important role in the function of flag leaves and starch synthetase activity during the reproductive stage [6,20]. Thus, the individual or combined treatment of the hormones improved rice growth parameters under salinity stress conditions.

Rice plants exhibit variable responses to stresses and hormonal treatments depending on the growth stage. At the booting and reproductive stages, rice plants show sensitivity to hormonal treatments such as ethylene, ABA, and BRs under raised salinity conditions, which affects the grain setting rate and spikelet development [6,12,29,30,33]. In addition, poor pollen viability and low yield under salinity stress occur in response to hormonal imbalance, which hinders soluble sugar translocation to spikelets and inhibits starch synthetase activity during spikelet development [33]. The results of this study are consistent with this finding. All the salinity levels reduced the pollen viability and caused spikelet sterility, whereas the exogenous application of BRs and ABA improved both of these qualities (Figure 4). Pollen viability and spikelet sterility can be a direct measure of the rice yield. In this regard, the rice 1000-grain weight and grain yield per plant were investigated under different salinity conditions. The treatments of ABA, BRs, and ABA+BRs improved the 1000-grain weight and grain yield per plant. These improvements in the 1000-grain weight and yield per plant are directly linked to the Pn and hormonal imbalance under salinity stress. The improvement in 1000-grain weight and yield per plant, after exogenous application of ABA and BRs under salinity stress, could be due to ethylene inhibition, which causes carbon assimilation, and therefore an improvement in photosynthesis [12,33]. Thus, the exogenous application of BRs and ABA could play a vital role in transducing the signals and in triggering the downstream responses conferring rice tolerance to salinity. Previous studies have shown that BRs increased growth and alleviated the deleterious effects induced by salinity stress in pea plants (*Pisum sativum* L.). Likewise, the treatment with 5 μM L^−1^ BRs detoxified the stress generated by NaCl and significantly improved growth, level of pigmentation, green pod yield, and pod protein in *Phaseolus vulgaris* L. [21,23]. These results provide strong evidence that ABA and BRs could be better options for improving rice growth and grain development under salinity stress. The findings from this study will help us to understand hormonal homeostasis in saline environments, which could help in designing better solutions to rescue rice crops in field conditions. The exogenous application of hormones, such as ABA and BRs, to combat ethylene production could be a better option for improving spikelet development in the future. Furthermore, the results suggest ways to resolve the problem of low rice grain yield, hormone homeostasis, and poor rice tolerance to salinity stress.

## 5. Conclusions

In this study, we evaluated the impact of ABA, BRs, and the interplay of ABA-BRs on rice plants under salt stress. The results revealed that the exogenous application of ABA, BRs, and ABA+BRs improved the agronomic parameters and photosynthesis characteristics in rice subjected to various salt stress levels. Application of BRs and ABA+BRs maintained the phytohormonal homeostasis, including ABA, BRs, JA, GA3, and IAA. Further, the exogenous application of hormones improved the Pn and SPAD value in rice flag leaf, improved the capacity of source-to-sink in spikelets, enhanced leaf area, better pollen viability, and increased 1000-grain weight and grain yield per plant under CK, LS, and HS salt stress levels.

In the future, the use of exogenous hormones, such as BRs and ABA, or their combination, can be used to improve plant growth and grain development by improving pollen viability and reducing spikelet sterility under saline conditions. On the other hand, the efficacy of such treatments at the field level and their cost-effectiveness should be investigated. Overall, the exogenous application of phytohormones can be explored in agricultural crops for enhancing the yield and resilience subjected to stresses.

## Figures and Tables

**Figure 1 life-13-00498-f001:**
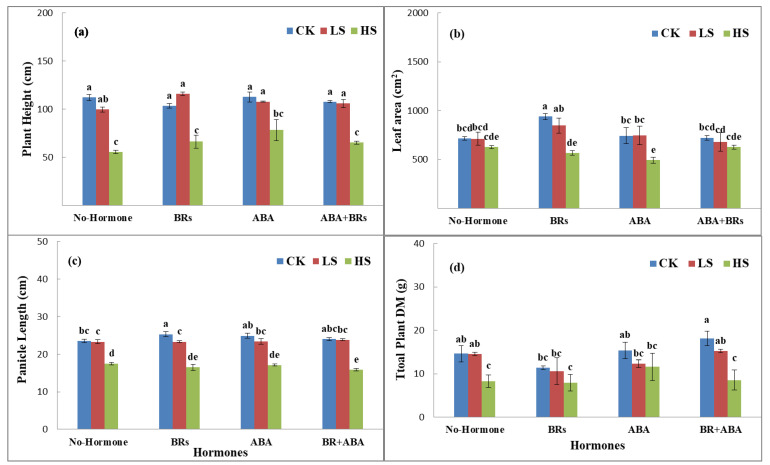
Effects of ABA and BRs on agronomic characteristics of rice under salt stress at full heading stage. (**a**) Plant height, (**b**) leaf area, (**c**) panicle length, and (**d**) total plant dry matter. The values are denoted as the mean ± SE (n = 3). Values followed by the same letters are not significantly different (P ≤ by the same letter LSD). ABA = abscisic acid; BRs = brassinosteroids. Similarly, CK, LS, and HS stand for control, low salt stress, and heavy salt stress, respectively. a, b, c, and d show the parameter number.

**Figure 2 life-13-00498-f002:**
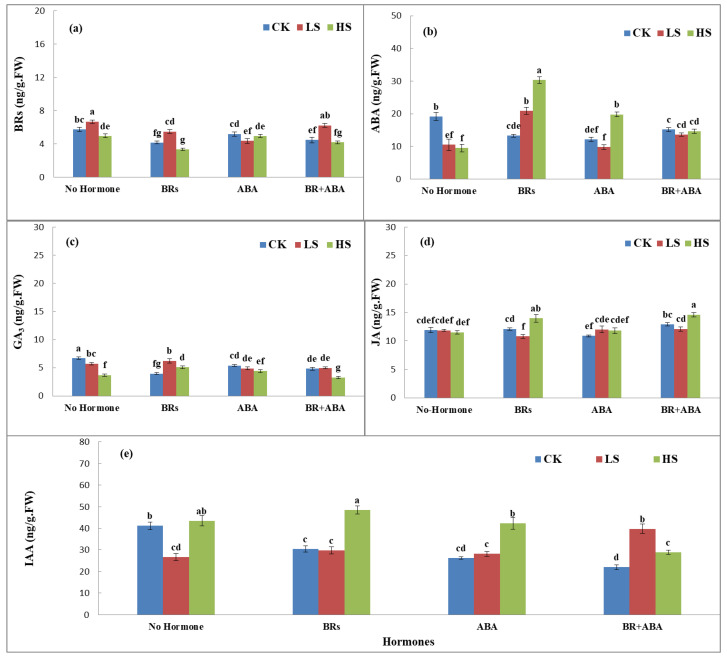
Effects of PLP and salt level on the production of (**a**) BRs, (**b**) ABA, (**c**) GA3, (**d**) JA, and (**e**) IAA hormones at the booting stage. The values are denoted as means. The hormones showed increased responses with BR treatment. However, differences were only significant with ABA+BR treatment, according to Tukey’s honestly significant difference (HSD) test. The treatments included control (CK), low salinity (LS), and high salinity (HS). ABA = abscisic acid; GA_3_ = gibberellic acid; BRs = brassinosteroids; JA = jasmonic acid; IAA = indole-3-acetic acid. The letters a, b, c, and d show the parameter number. Similarly, CK, LS, and HS stand for control, low salt stress, and heavy salt stress, respectively.

**Figure 3 life-13-00498-f003:**
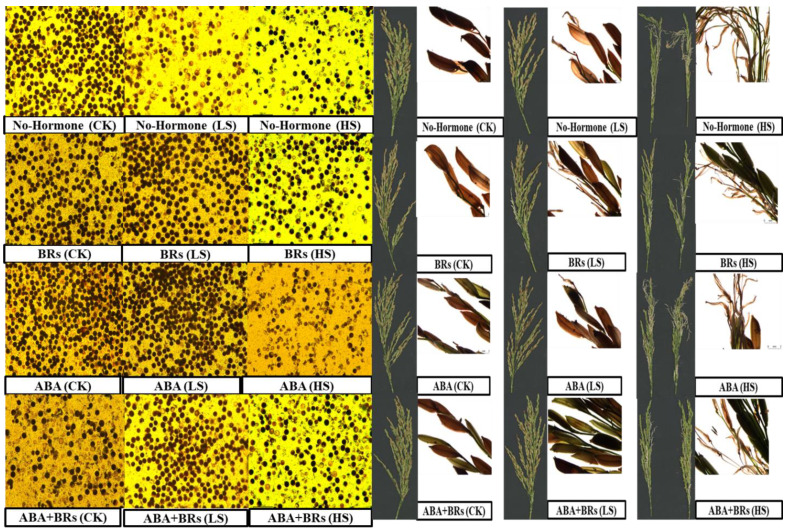
Examination of pollen viability and spikelet sterility at the anthesis stage on rice cultivars subjected to exogenous hormones at different salt stress levels. ABA = abscisic acid; BRs = brassinosteroids; CK = no salt stress (control); LS = (low salt stress); HS = heavy salt stress.

**Figure 4 life-13-00498-f004:**
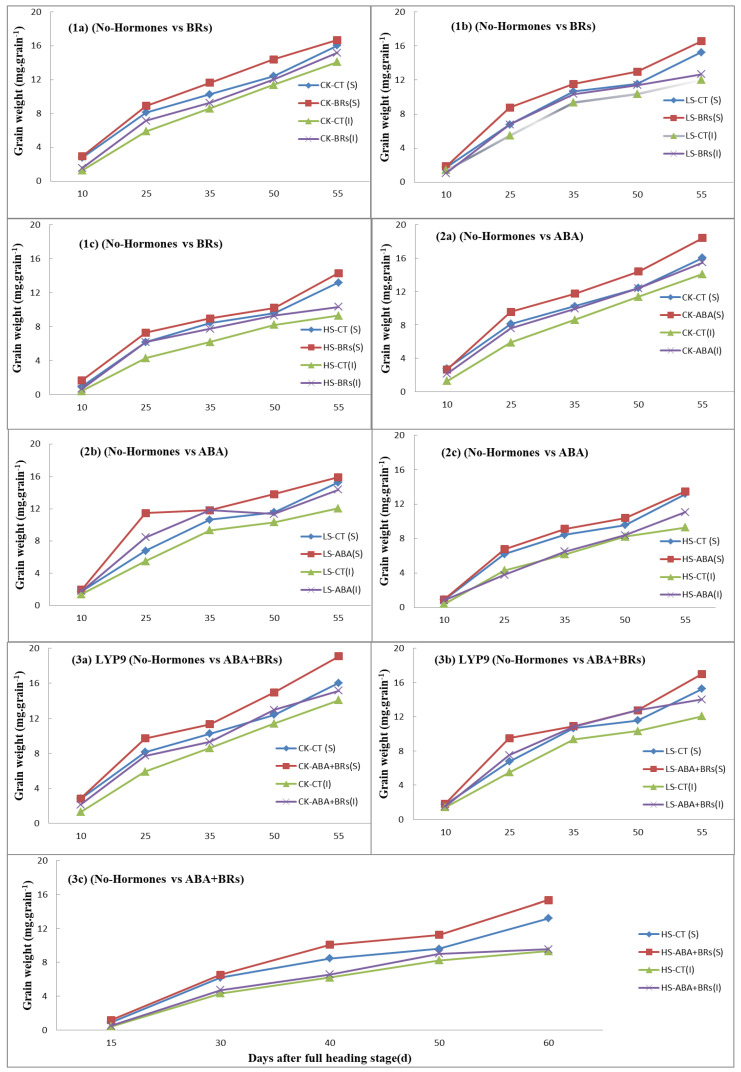
Effects of exogenous hormones on grain development (grain weight) with respect to time in LYP9 under salt stress. Effects of application of BRs (**1a**–**1c**) and ABA (**2a**–**2c**), and combined effect of ABA+BRs (**3a**–**3c**) on grain weight compared to control (no hormones). ABA = abscisic acid; BRs = brassinosteroids; CK = no salt stress (control); LS = low salt stress; HS = heavy salt stress; CT = no hormone; S = superior spikelets; I = inferior spikelets. a = CK; b = LS; and c = HS. Similarly, CK, LS, and HS stand for control, low salt stress, and heavy salt stress, respectively.

**Table 1 life-13-00498-t001:** Crop data 2019.

Rice Cultivar	Sowing Date dd/mm	Seedlings Per Pot	Transplanting Stage dd/mm	Maximum Tillering dd/mm	Full Heading dd/mm	Maturity dd/mm
Liangyoupiejiu	30/05	6 Seedlings	30/06	06/08	23/08	25/10

**Table 2 life-13-00498-t002:** Response to hormone application of photosynthesis attributes under salinity stress.

Hormones Levels	Salt Levels	Net Photosynthesis Rate (µmol CO_2_ m^−2^ s^−1^)	Stomatal Conductance (mol m^−2^ s^−1^)	Transpiration Rate (mmol m^−2^ s^−1^)	SPAD Value
no hormone	CK	28.67 ± 2.19 abc	0.82 ± 0.01 a	88.0 ± 1.52 a	41.4 ± 0.25 abc
	LS	25.67 ± 0.88 bcd	0.62 ± 0.14 abc	73.7 ± 8.7 abc	38.0 ± 0.7 cd
	HS	23.0 ± 1.53 cd	0.33 ± 0.07 cd	54.3 ± 8.6 bcd	37.0 ± 0.8 d
BRs	CK	28.67 ± 0.88 abc	0.66 ± 0.09 abc	81.7 ± 1.3 ab	40.2 ± 0.2 abcd
	LS	23.67 ± 0.67 cd	0.38 ± 0.1 bcd	60.7 ± 8.7 abcd	40.2 ± 0.5 abcd
	HS	22.33 ± 0.88 cd	0.27 ± 0.04 d	47.7 ± 3.0 cd	43.3 ± 0.7 a
ABA	CK	33.0 ± 2.08 a	0.92 ± 0.03 a	87.0 ± 1.2 a	41.8 ± 1.3 ab
	LS	25.0 ± 1.15 bcd	0.44 ± 0.03 bcd	64.7 ± 2.2 abcd	40.0 ± 0.5 abcd
	HS	21.0 ± 1.0 d	0.43 ± 0.02 bcd	56.0 ± 8.5 bcd	40.2 ± 0.8 abcd
ABA+BRs	CK	32.0 ± 2.08 ab	0.73 ± 0.07 ab	81.7 ± 3.8 ab	42.0 ± 0.5 ab
	LS	25.86 ± 0.71 bc	0.52 ± 0.04 bc	66.6 ± 2.8 abc	40.6 ± 0.4 abc
	HS	23.67 ± 1.33 cd	0.43 ± 0.05 bcd	62.3 ± 2.0 abcd	40.3 ± 0.3 abc

Values are denoted as mean ± SE and lettering a, b, c, and d shows the pair mean difference. ABA = abscisic acid; BRs = brassinosteroids, Similarly, CK, LS, and HS stand for control, low salt stress, and heavy salt stress, respectively.

**Table 3 life-13-00498-t003:** Effect of exogenous hormones on yield and its components.

Treatments	Salt Levels	1000-Grain Weight (g)	Yield Per Plant (g)	Seed Setting Rate (%)	Total Above-Ground Biomass Per Plant (g)
no hormone	CK	25.67 ± 0.17 a	25.47 ± 1.26 a	59.2 ± 0.21 ab	244.45 ± 20.9 ab
	LS	24.54 ± 0.16 a	23.9 ± 0.89 a	57.2 ± 0.50 ab	223.76 ± 8.18 b
	HS	19.33 ± 0.19 b	5.33 ± 0.85 b	43.0 ± 0.74 b	122.21 ± 16.6 d
BRs	CK	25.10 ± 0.61 a	25.50 ± 0.42 a	67.8 ± 0.35 a	238.5 ± 13.6 ab
	LS	23.89 ± 0. 17 a	25.40 ± 4.5 a	57.7 ± 1.14 ab	245.2 ± 24.1 ab
	HS	20.77 ± 0.85 b	9.40 ± 1.42 b	53.2 ± 0.66 ab	177.3 ± 9.9 c
ABA	CK	26.21 ± 0.43 a	22.37 ± 0.55 a	48.7 ± 0.11 ab	258.7 ± 21.4 ab
	LS	24.98 ± 0.69 a	20.97 ± 0.72 a	53.7 ± 0.54 ab	231.2 ± 8.6 ab
	HS	19.09 ± 1.74 b	8.13 ± 0.38 b	49.7 ± 0.46 ab	136.06 ± 5.2 cd
ABA+BRs	CK	25.59 ± 0.18 a	23.40 ± 0.35 a	46.9 ± 032 b	273.0 ± 14.6 a
	LS	24.86 ± 0.22 a	22.50 ± 0.75 a	41.6 ± 0.94 b	271.1 ± 2.3 ab
	HS	16.58 ± 0.70 c	6.0 ± 0.06 b	39.8 ± 0.40 b	135.0 ± 11.8 cd

Values are denoted as mean ± SE and lettering a, b, c, and d shows the pair mean difference. ABA= abscisic acid; BRs = brassinosteroids, Similarly, CK, LS, and HS stand for control, low salt stress, and heavy salt stress, respectively.

## Data Availability

Data will be made available with a reasonable request to the corresponding authors.

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
