# Peer review of "Interplay Impact of Exogenous Application of Abscisic Acid (ABA) and Brassinosteroids (BRs) in Rice Growth, Physiology, and Resistance under Sodium Chloride Stress"

_life, 2023, doi:10.3390/life13020498_

Round 1

Reviewer 1 Report (New Reviewer)

The study on the role of ABA and BRs on enhancing tolerance to salinity stress is relevant. However, the language of the manuscript needs extensive editing to convey the intended meaning.

Author Response

Response to Reviewer 1 Comments

Point 1: The study on the role of ABA and BRs on enhancing tolerance to salinity stress is relevant. However, the language of the manuscript needs extensive editing to convey the intended meaning.

Response 1: Thank you for the appreciation. To improve the English of the manuscript, we have used the MDPI English editing service. The English Language Editing certificate is attached as a reference. We hope now the manuscript will be better readable.

Reviewer 2 Report (New Reviewer)

Dear authors,

[For Introduction]

1. Please considered to include whether previous studies reported the crosstalk between ABA and BR signaling in rice since dual hormone treatment was tested in this manuscript.

2. Exogenous hormone application to mitigate negative effect from high salinity condition has been tested by other group, and this study focused on the synergistic effects of  the ABA and BR on salinity stress tolerance of rice. Is this strategy cost-effective in financial consideration to improve agronomic trait of corresponding cultivar?

[Materials and Methods]

3. In 2.2. Treatment Plan section, please considered to spell out specific term. Ex. EC, electrical conductivity?

4. In 2.3 Hormone application section, the data of synergistic group  (ABA+BRs) was shown or not?

5. In 2.6 Photosynthesis parameters and leaf area section, the duration of day light between 8:30 am to 11:30 pm (or am)?

6. In 2.7 Quantification of the hormones produced in rice flag leaf section, please considered to record the centrifuge speed in RCF unit (g), or in RPM unit but with information of rotor and centrifuge machine.

[For Results]

7. In 3.1 section, please check the description of plant height after BR treatment among different experimental groups again.

8. In Figure 1, the plant height in LS might be higher than CK and the panicle length in LS was significantly shorter than CK in BR group. Is this imply the BR treatment enhance growth of plant and inhibit panicle growth under 2 dS/m specifically. What would be possible explanation that HS did not phenocopies LS in presence of exogenous BR application?

9. In Figure 3, please considered to show pollen viability in white background.  As displayed in current manuscript, yellow background did not provide better resolution to distinguish pollen colors for viability. Together with the images of spikelets sterility, please provide qualitative results for pollen viability for better comparison among different experimental group.

10. In Figure 4, please considered to provide statistical analysis for line chart among different groups. 

11. What is the agent to induce salinity stress? NaCl, KCl or other chemicals? 

Author Response

Response to Reviewer 2 Comments

Point 1: [For Introduction]

  1. Please considered to include whether previous studies reported the crosstalk between ABA and BR signaling in rice since dual hormone treatment was tested in this manuscript.

Response 1: Thank you for your suggestion. We have added a few lines into the introduction on the crosstalk of ABA and BR in rice.

Point 2: Exogenous hormone application to mitigate negative effect from high salinity condition has been tested by other group, and this study focused on the synergistic effects of  the ABA and BR on salinity stress tolerance of rice. Is this strategy cost-effective in financial consideration to improve agronomic trait of corresponding cultivar?

Response 2: This is an excellent question. Unfortunately, we are unable to provide the confirmed answer as the economic side of the experiment has not yet been investigated. Additionally, multi-season and complete filed evaluations have not been done. The current research was more confined to a lab scale and to find out the effects of ABA and BR treatments in rice under salt stress.

Point 3: [Materials and Methods]

In 2.2. Treatment Plan section, please considered to spell out specific term. Ex. EC, electrical conductivity?

Response 3: The word EC stand for Electrical Conductivity. This word has been added in the manuscript. Changes are hilighted in red.

Point 4: [Materials and Methods]

. In 2.3 Hormone application section, the data of synergistic group (ABA+BRs) was shown or not?

Response 4: The synergistic group (ABA+BRs) data of pre-experiment is not shown in the manuscript. However, I am sharing pre-experiment figures.

Point 5: [Materials and Methods]

In 2.6 Photosynthesis parameters and leaf area section, the duration of day light between 8:30 am to 11:30 pm (or am)?

Response 5: Please accept our apology for the typo error. The photosynthesis paramters were taken in the day light between 8:30 am to 11:30 am. The correction has been made in the revised manuscript.

Point 6: [Materials and Methods]

In 2.7 Quantification of the hormones produced in rice flag leaf section, please considered to record the centrifuge speed in RCF unit (g), or in RPM unit but with information of rotor and centrifuge machine.

Response 6:

The sample was centrifuged with 3500 rpm for 8 min centrifuged machine having high performance fixed angle rotor.

Point 7: [For Results]

In 3.1 section, please check the description of plant height after BR treatment among different experimental groups again.

Response 7:  In this section, BRs showed better results but non-significance in terms of plant height compared to CK. 

Point 8: [For Results]

  1. In Figure 1, the plant height in LS might be higher than CK and the panicle length in LS was significantly shorter than CK in BR group. Is this imply the BR treatment enhance growth of plant and inhibit panicle growth under 2 dS/m specifically. What would be possible explanation that HS did not phenocopies LS in presence of exogenous BR application?

Response 8:  It is very important question. The results in term of plant height showed better results than CK but non-signficant. BRs shows improvement in plant height and less in penicle length this could be due to response of BRs on growth stages. However, I feel sorry that I am unable to answer this question fully.

Point 9: [For Results]

In Figure 3, please considered to show pollen viability in white background. As displayed in current manuscript, yellow background did not provide better resolution to distinguish pollen colors for viability. Together with the images of spikelets sterility, please provide qualitative results for pollen viability for better comparison among different experimental group.

Response 9: These figures are original figures with yellow color due to potassium iodide (KI). The black pollens clearly showing the fertile or viabale. Please consider this figure in current form.

Original Images:

With change background:

Point 10: [For Results]

In Figure 4, please considered to provide statistical analysis for line chart among different groups.

Response 10:  Dear Sir, This figure is about the grain developemnt with respact to time. As per my understading the possible way to represent the data is in linear graph. Please consider it.

Point 11: [For Results]

What is the agent to induce salinity stress? NaCl, KCl or other chemicals?.

Response 11: The salinity stress was induced by using NaCl salt. The correction has been made in the revised manuscript.

Reviewer 3 Report (New Reviewer)

The Research work titled "Exogenous Application of Abscisic Acid (ABA) and Brassinosteroids (BRs) Revealed Their Roles in Rice Growth, Physiology, and Resistance under Sodium Chloride Stress" is a niece piece of work and can be conisdered for publication after necessary revsison.

1. Title of the manuscript need to be modified, like "Interplay Impact of Exogenous Application of Abscisic Acid (ABA) and Brassinosteroids (BRs)  in Rice Growth, Physiology, and Resistance under Sodium Chloride Stress" 

2. Figure legends should be provided with all the information like what what CK, LS and  and HS denotes in Figure 1.

3.The overall graphs of the manuscript should be improved, so that the readers can understand the story easily  (Figure 2, Figure 4 etc)

4. The methods for Quantification of the hormones produced in rice flag leaf need to be eloborated more.

5. The authors should also provide data regarding the impact of about the metabolites and phytochemical changes if available

Author Response

Response to Reviewer 3 Comments

The Research work titled "Exogenous Application of Abscisic Acid (ABA) and Brassinosteroids (BRs) Revealed Their Roles in Rice Growth, Physiology, and Resistance under Sodium Chloride Stress" is a niece piece of work and can be conisdered for publication after necessary revsison.

Response: Thank you so much for your comments and appreciation.

Point 1: The title of the manuscript needs to be modified, like "Interplay Impact of Exogenous Application of Abscisic Acid (ABA) and Brassinosteroids (BRs)  in Rice Growth, Physiology, and Resistance under Sodium Chloride Stress".

Response 1: Thank you for the valuable suggestion. The manuscript title has been modified accordingly.

Point 2: Figure legends should be provided with all the information like what what CK, LS and  and HS denotes in Figure 1.

Response 2: The detail of CK, LS, and HS has been added in the figure 1 legend.

Point 3: The overall graphs of the manuscript should be improved, so that the readers can understand the story easily  (Figure 2, Figure 4 etc)

Response 3: We have modified the figure legends to improve the readability. We hope now they are easy to follow.

Point 4: The methods for Quantification of the hormones produced in rice flag leaf need to be elaborated more.

Response 4:

The method of quantification of hormones has been elaborated in the revised manuscript.

Point 5: The authors should also provide data regarding the impact of about the metabolites and phytochemical changes if available.

Response 5: Thanks for the suggestion. Unfortunately, we have not performed any metabolomics studies on this. Therefore, no data is with us about the phytochemicals and metabolic dynamics. However, we will keep this suggestion in mind for our subsequent planned experiemnts.

Round 2

Reviewer 1 Report (New Reviewer)

The methodology part and minor corrections made in the manuscript need to be changed.

Author Response

Point 1: The methodology part and minor corrections made in the manuscript need to be changed.

Response 1: Thank you for your valuable time and review of this manuscript.

The correction has been made in the revised manuscript and rephrases the paragraph mentioned in the methodology.

Reviewer 2 Report (New Reviewer)

Dear Authors,

I am satisfied with current revision and do not have further comments on your manuscript, thank you.

Author Response

Thank you for your time and appreciation for this manuscript. Your suggestions have improved the quality of the manuscript.  

This manuscript is a resubmission of an earlier submission. The following is a list of the peer review reports and author responses from that submission.

Round 1

Reviewer 1 Report

The authors response as shared by Mr. Zhang was not satisfactory. Most of the suggested changes were ignored. 

Reviewer 2 Report

The authors investigated the effects of exogenous application of plant phytohormone on rice spikelet development subjected salinity stress. It is a very meaningful study. However, the manuscript didn’t investigate the associated mechanism.

  According to Table 2, Pn of plant applicate with BR in HS is 22.33 22.33±0.88cd, applicate with ABA in HS is 21.0±1.0d, respectively. Both are decreased compared with the plant with no hormone applicate in HS(23.0±1.53cd). while in line 221-223, the authors described that Pn was increased with BRs, ABA and ABA+BRs treatment compared to CK in high salt stress?